# Purinergic-Mediated Calcium Signaling in Quiescent and Activated Hepatic Stellate Cells: Evidence That P2Y1 Receptor Delays Activation

**DOI:** 10.3390/cells14231845

**Published:** 2025-11-23

**Authors:** Esperanza Mata-Martínez, Ana Patricia Juárez-Mercado, Adriana González-Gallardo, José David Núñez-Ríos, Mauricio Díaz-Muñoz, Rolando Hernández-Muñoz, Francisco G. Vázquez-Cuevas

**Affiliations:** 1Departamento de Biología Celular y Desarrollo, Instituto de Fisiología Celular, Universidad Nacional Autónoma de México, Mexico City CP 04510, Mexico; espemmtz@gmail.com; 2Departamento de Neurobiología Celular y Molecular, Instituto de Neurobiología, Universidad Nacional Autónoma de México, Querétaro CP 76230, Mexico; apjm0990@gmail.com (A.P.J.-M.); jd.nunezrios@gmail.com (J.D.N.-R.); mdiaz@comunidad.unam.mx (M.D.-M.); 3Unidad de Proteogenómica, Instituto de Neurobiología, Universidad Nacional Autónoma de México, Querétaro CP 76230, Mexico; gallardog@unam.mx

**Keywords:** purinergic receptors, phenotypic transdifferentiation, adenine/uridine nucleotides, intracellular calcium dynamics, hepatic stellate cells activation, P2Y1 receptor

## Abstract

**Highlights:**

**What are the main findings?**
A variety of Gq-coupled P2Y receptors are functional in qHSC and MFB.P2Y1 receptor is first described in qHSC and its stimulation regulates the transdifferentiation process.

**What is the implication of the main finding?**
The study of purinergic signaling in HSC has great potential to gain more understanding about liver fibrogenesis.Specifically, P2Y1 receptor might be a molecular target to fibrosis onset.

**Abstract:**

Hepatic stellate cells (HSC) play a crucial role in the fibrotic response of the liver when they transdifferentiate from quiescent cells (qHSC) to myofibroblast (MFB). Ca^2+^ responses mediated by purinergic P2Y receptors are not fully characterized in qHSC and MFB. The objective of this study was to compare the expression of purinergic receptors with the capacity to mobilize intracellular Ca^2+^ in both phenotypes, as well as to explore the potential role of these signals in HSCs activation. Isolated mouse HSC were quiescent on day 2 and became MFB on day 7 when cultured in high stiffness substrate. Both phenotypes expressed the transcripts of *P2ry1*, *P2ry2*, *P2ry6* and *P2x7*, and exhibited a similar Ca^2+^ response to UDP, UTP and Bz-ATP, indicating comparable activity in P2Y6, P2Y2 and P2X7 receptors. In contrast, *P2y12* transcript was detected only in MFB. Remarkably, P2Y1 receptor was identified in qHSC, an observation that had not yet been reported. Evidence of P2Y1 receptor functionality was obtained from stimulation with ADP. ADP-elicited Ca^2+^ mobilization was more potent in qHSC in comparison to MFB. Interestingly, ADP stimulation worsens the transdifferentiation of qHSC to MFB after 4 or 7 days in culture, strongly suggesting the role of this purinergic receptor in HSC activation.

## 1. Introduction

Hepatic stellate cells (HSC) are non-parenchymal cells of the liver classified as pericytes. These cells are located in the Disse space, a region between the sinusoidal endothelium and the parenchyma of the hepatic lobule [1]. HSC constitute around 10% of the total population of liver cells [1,2]. HSC are biomedically important because of their role in fibrosis onset and scarring. In the healthy liver, HSC exhibit a quiescent phenotype (qHSC) characterized by the presence of retinoid droplets; qHSC are central in the formation of the basal extracellular matrix, antigen presentation, and the production of growth factors and cytokines [1]. When the organ suffers a chronic injury, parenchymal and endothelial cells release various intercellular messengers that target HSC, inducing their transdifferentiation to myofibroblasts (MFB) this process is known as *activation* [3]. MFB produces collagen I-alpha, a hallmark of liver fibrosis. qHSC express peroxisome proliferator-activated receptor gamma and glial fibrillary acidic protein, whereas MFB express alpha-smooth muscle actin (αSMA) and collagen 1a1 (COL1A1) and lack retinoid deposits. In liver fibrosis, MFB mainly originates from qHSC [3,4]. Interestingly, activation can occur spontaneously, driven by biophysical conditions, when HSC are cultured in plastic Petri dishes for five to seven days [1,5]. Research has demonstrated that HSC activation can be reverted. In a model of hepatocellular injury by CCl_4_ administration, the removal of the toxic stimulus downregulates fibrotic genes, inhibits cell death and reverts the MFB phenotype [6]. Therefore, a comprehensive understanding of the mechanisms governing HSC activation is fundamental to better understanding their basic biology and for the development of innovative therapies.

Previous studies had explored purinergic responses in HSC. A pivotal study by Takemura and collaborators (1994) found that in rat, before the cloning of purinergic receptors, HSC stimulation by purinergic agonists (ATP, ADP and UTP) caused an increase in inositol triphosphate (IP_3_) and intracellular Ca^2+^ levels [7]. In 2004, JA Dranoff and colleagues published the first report characterizing purinergic receptor expression by analyzing transcript levels through reverse transcription and polymerase chain reaction (RT-PCR) and assessing function by intracellular calcium imaging. The authors concluded that qHSC express *P2ry2* and *P2ry4* transcripts, but both receptors were downregulated due to the activation process, whereas in MFB, *P2ry1* and *P2ry6* transcripts were upregulated after seven days in culture. They also noted that MFB expressed an unidentified purinoceptor. In functional experiments, this group observed that P2Y6 receptor stimulation in MFB induced an increment in procollagen-2 expression, suggesting a pro-fibrotic role for this receptor [8].

Acetaldehyde administration to rat qHSC for 48 h induced an increase in P2X7 receptor levels along with an elevation in typical fibrosis markers such as αSMA and COL1A1. Furthermore, activating the P2X7 receptor with the selective agonist 2′(3′)-O-(4-Benzoylbenzoyl) adenosine-5′-triphosphate (BzATP), was found to enhance the effects of acetaldehyde on proliferation, inflammation, and the expression of activation markers. Acetaldehyde-mediated activation of HSC by the P2X7 receptor was abolished by genetic suppression and pharmacological inhibition with the selective antagonist A438079 [9]. On the other hand, hepatocellular damage in mice also induced overexpression of the P2X4 receptor; a similar upregulation was observed when the HSC-T6 cell line was exposed to acetaldehyde. Furthermore, incubation of the HSC-T6 line with the selective antagonist of the P2X4 receptor 5-(3-Bromophenyl)-1,3-dihydro-2*H*-benzofuro [3,2-*e*]-1,4-diazepin-2-one (5-BDBD), blocked the acetaldehyde-dependent activation of this cell line [10].

An elegant study evaluating the contribution of injured parenchymal cells to HSC activation by damage-associated molecular patterns (DAMPs) and their receptors revealed that the UDP-glucose/UDP-galactose pair acted as ligands for the P2Y14 receptor in mice. Co-culture of damaged hepatocytes or incubation with a P2Y14 receptor agonist induced the appearance of MFB. Furthermore, total or HSC-directed knockout of P2Y14 receptor reduced fibrosis induction in different hepatotoxic models [11].

Altogether, evidence demonstrates that purinergic receptors participate in processes associated with the induction of liver fibrosis, specifically in HSC activation. However, molecular mechanisms remain poorly understood. This study systematically compares purinergic responses in two distinct HSC phenotypes (qHSC and MFB) by analyzing Ca^2+^ transients induced by various physiological and pharmacological agents. Unexpectedly, our data provides the first functional and pharmacological evidence of P2Y1 receptor expression in liver qHSC.

## 2. Materials and Methods

### 2.1. Animal Care and Use Statement

The animal management protocol was designed to inflict as little pain as possible. Animals were kept under control conditions (23 °C, a 12/12 h light–dark cycle, and food and water ad libitum). All experimental procedures were approved by the Institute of Neurobiology’s Bioethics Committee at the National Autonomous University of México (UNAM) (protocol 85.A) on 29 April 2015 and comply with Official Mexican Standard SAGARPA NOM-062-ZOO-1999 [12].

### 2.2. Hepatic Stellate Cells Isolation

HSC were isolated following previously published protocols [13]. Briefly, C57BL mice aged 24 to 32 weeks were anesthetized, and a laparotomy was performed to expose the portal vein and inferior vena cava around the liver. Once the portal vein was cannulated with a catheter, the organ was perfused with EGTA solution (in mM: 137 NaCl, 5 KCl, 0.5 NaH_2_PO_4_*H_2_O, 0.8 Na_2_HPO_4_, 9.9 HEPES, 4.1 NaHCO_3_, 0.5 EGTA, 5 Glucose). This was followed by in situ digestion using pronase (242 μg/mL) and then collagenase (85 μg/mL) solutions. After, the minced liver was further digested in vitro with pronase plus collagenase solution (250 μg/mL and 86 μg/mL, respectively). Finally, HSC were separated from other hepatic cell populations by density gradient centrifugation in Histodenz solution (29%). After collection, HSC were cultured in DMEM High Glucose complemented with 10% fetal bovine serum and antibiotic–antimycotic on Petri dishes or coverslips at 37 °C and maintained in a humidified CO_2_ incubator. Transdifferentiation led to the appearance of MFB after 7 days of culture on a rigid plastic surface [14].

### 2.3. Reverse Transcription and Polymerase Chain Reaction

Markers for qHSC and MFB, as well as P2Y receptor expression were analyzed by reverse transcription polymerase chain reaction (RT-PCR). Total RNA was isolated with Trizol reagent (Thermo Scientific, MA, USA) according to the manufacturer’s protocol. RT reaction was performed using 1 μg of total RNA, oligo dT, and 1 U of M-MLV reverse transcriptase (GoScript, PROMEGA, WI, USA) and RNase inhibitor (RNasin, PROMEGA, WI, USA). All amplification protocols started and finished with a hold of 5 min at 94 °C and 72 °C, respectively. The oligonucleotides utilized are listed in Table 1.

### 2.4. Intracellular Calcium Measurement by Fluorescent Microscopy

This procedure was previously reported [15]. Briefly, isolated qHSC and MFB were placed onto coverslips and incubated at 37 °C for 30 min with Fluo-4 AM 2 μM (Thermo Scientific, Waltham, MA, USA) in Krebs solution: 150 NaCl, 1 KCl, 1.5 CaCl_2_, 1 MgCl_2_, 10 HEPES, and 4 glucose (all in mM), with pH adjusted to 7.4 in the presence of 5% CO_2_ and 95% O_2_. Unincorporated dye was washed for removal. The calcium-associated fluorescent signal was visualized with a Nikon Eclipse Ts2R-FL inverted microscope. Images from the recorded videos were collected every 500 ms and acquired with a 20X microscope objective coupled to a Retiga Electro CCD camera with Ocular scientific image acquisition software 2.0 (Teledyne Photometrics, Tucson, AR, USA). A pipette was used to manually apply the purinergic ligands. When inhibitors were used, they were pre-incubated for 5 to 15 min (according to the drug action) before the upcoming stimulus. The recordings were performed in Normal-Ca^2+^ (NCa^2+^): [Ca^2+^] = 1.5 mM or in Zero-Ca^2+^ (0 Ca^2+^), where CaCl_2_ was practically omitted by the addition of 2.36 mM EGTA: [Ca^2+^] = 24 pM; Maxchelator v8 (Webmaxc standard; UC, Davis).

Fluo-4 in cellular samples was excited at 488 nm, and the emission signal was recorded at 525 nm. Unprocessed fluorescence intensity values were analyzed with Image J version Fiji (National Institutes of Health, USA) and normalized using the equation ((F/F0) − 1)(100))/(Fmax), where F = fluorescence intensity measured at any given time, F0 = minimum fluorescence intensity obtained before the addition of any stimulus, and Fmax = maximum fluorescence intensity obtained after adding 10 µM of ionomycin; these values were plotted vs. time. Total quenching of the calcium signal was achieved by the addition of 5 mM of MnCl_2_.

### 2.5. Immunofluorescence

Immunofluorescence labeling of qHSC or MFB was performed according to previously published protocols [16]. In brief, cell cultures were fixed in 4% paraformaldehyde for 20 min; then, the samples were permeabilized by incubation in 1% Triton X-100 and blocked with 5% fat-free milk. All reactives were diluted in phosphate-buffered saline (PBS; in mM: NaCl 136, KCl 2.7, Na_2_HPO_4_ 10, KH_2_PO_4_ 1.8, pH 7.4). The primary antibodies, anti-α-SMA (#19245 Cell Signaling, Danvers, MA, USA) and anti-YAP (sc-101199, Santa Cruz Biotechnology, Dallas, TX, USA), were used at a dilution of 1:100, and the secondary antibody, goat anti-rabbit IgG-Alexa Fluor 488, was applied at a 1:200 dilution (Invitrogen, Waltham, MA, USA). When indicated, phalloidin conjugated with Alexa Fluor 635, DAPI, or propidium iodide (ThermoFisher Scientific, Waltham, MA, USA) were added during the final washes after secondary antibodies, at the concentration recommended by the manufacturer. Samples were mounted in Vectashield (Vector Labs, Newark, CA, USA) and observed under an LSM 780 confocal microscope (Karl Zeiss, Oberkochen, Germany).

### 2.6. Analysis of P2RY1 Transcript Expression in Public Databases

Gene Expression Omnibus [17] was utilized to find datasets containing human HSC, both quiescent and activated. Subsequently, the GEO2R tool [18] was employed to analyze gene expression data from MFBs and compare them to qHSC. The Benjamini and Hochberg (False Discovery Rate) option was applied for *p*-value adjustment (significant adjusted *p*-value < 0.05). Datasets were selected from healthy human donors with HSCs that had been activated with TGFβ at various concentrations and stimulation timings. The study focused on P2Y1 receptor transcripts. Table 2 summarizes the experimental conditions and their corresponding references.

### 2.7. Biostatistics

Gaussian distribution of the data was tested with D’Agostino and Pearson omnibus normality and the Shapiro–Wilk normality test. Statistical differences among treatments and conditions were assessed using Mann–Whitney test, Kruskal–Wallis test. A probability (*p*) value < 0.05 was considered a significant difference. The statistical analysis was performed using GraphPad Prism version 5.01 (GraphPad). Additionally, Dr. Nuri Aranda-López, professor of statistical methodologies at UNAM, reviewed the statistical analysis.

## 3. Results

### 3.1. Primary Cultured qHSC Transdifferentiate to MFB

Isolated mouse qHSC displayed their typical phenotype in culture (Figure 1A, left), characterized by a stellate shape and the presence of multiple retinol vesicles within the cytoplasm (Figure 1A right, orange arrowheads). On the second day in culture, HSC were identified as quiescent (qHSC), whereas after seven days, they transdifferentiated to an MFB phenotype [14]. MFB transcript markers *Col1A*, encoding for collagen 1, and *Acta2*, encoding for αSMA, were detected using RT-PCR. MFB exhibited increased expression levels of the two transcripts (Figure 1B). *Sod2* (superoxide dismutase 2) was used as a housekeeping transcript. Immunofluorescence analysis revealed discreet expression of αSMA in qHSC but abundant expression in MFB (Figure 1C). The MFB phenotype was markedly different, characterized by a large area of αSMA-positive extracellular matrix (Figure 1C). Notably, the 480 nm blue laser produced a yellow tint in the retinol vesicles (Figure 1C, left).

### 3.2. MFB and qHSC Express P2Y Receptors

This study focused on examining P2Y receptors coupled to intracellular Ca^2+^ mobilization. First, we analyzed the expression of *P2ry1*, *P2ry2*, *P2ry4*, and *P2ry6* transcripts by RT-PCR. The P2Y11 receptor, while coupled to Gq proteins, was not analyzed because it is not expressed in rodents [25]. Additionally, we identified the *P2ry12* transcript, which is generally coupled to Gi proteins. As a control for amplification, liver homogenate cDNA was used as a template. The reactions confirmed that the amplicon sizes were as expected: *P2ry1*, 315 bp; *P2ry2*, 174 bp; *P2ry6*, 152 bp; *P2ry12*, 162 bp; *P2ry7*, 129 bp; *Sod2*, 77 bp (Figure 2, lower panel).

In qHSC, we identified the expression of *P2ry1*, *P2ry2*, *P2ry6*, and *P2rx7* transcripts; however (Figure 2, upper panel), the *P2ry4* transcript was not detected. A similar expression pattern was observed in MFB, except *P2ry12* was also clearly detected (Figure 2, middle panel). Amplicons were purified and sequenced, followed by analysis using the BLAST 2.17.0 nucleotide platform [26]. The analysis confirmed that *P2ry1* corresponded to NM_008772.5, *P2ry2* to NM_008773.4, *P2ry6* to NM_183168, *P2ry12* to NM_027571.4, *P2rx7* to NM_011027.4, and *Sod2* to NM_013671.3.

### 3.3. Calcium Responses Elicited by Purinergic Agonists in qHSC and MFB

Calcium mobilization elicited by purinergic receptors was assessed by functional microscopy using Fluo-4 AM dye, according to previously published protocols [15]. First, we identified the presence of a “total” purinergic response by stimulating with 100 μM of ATP to activate all the P2Y and P2X7 receptors. Figure 3A presents a representative series of images illustrating a response elicited by 100 μM of ATP in qHSC and MFB in NCa^2+^ conditions. The fluorescence signal is shown in a pseudocolor scale. The recordings initially revealed the effect of the purinergic ligand, followed by ionomycin and MnCl_2_, to determine the maximum and minimal Ca^2+^ levels, as outlined in the methods. Under normal extracellular calcium conditions (NCa^2+^), ATP induced a potent Ca^2+^ response in qHSC that reached a peak within milliseconds and was sustained for at least 2.5 min with a slow decline of ~20% over this lapse (Figure 3B). A response exhibiting similar kinetic but minor amplitude was observed in MFB (130.2 ± 7.2 vs. 112.9 ± 10.9 area under the curve [AUC] for qHSC and MFB, respectively; Figure 3C). When the recordings were performed in 0 Ca^2+^ extracellular solution, qHSC and MFB presented an initial Ca^2+^ peak with a similar magnitude to those observed in NCa^2+^. However, the cation quickly returned to basal levels (~120 s) in both cell types, suggesting that the response elicited by 100 μM ATP in NCa^2+^ had an influx component that supported a prolonged elevation of cytosolic Ca^2+^.

Responses induced by a group of P2Y and P2X7 agonists with distinct selectivity were used to examine nucleotide-mediated Ca^2+^ mobilization in qHSC and MFB. The responses elicited by UTP in NCa^2+^ exhibited a fast increment that gradually returned to the basal level, with a more pronounced slope of decrease observed in qHSC than in MFB (Figure 4A, left). In 0 Ca^2+^ extracellular solution, the response elicited by UTP in both phenotypes consisted of a rapid peak that returned to baseline ~1.5 min. The primary difference we detected was the slope of responses in MFB under NCa^2+^ conditions compared to 0 Ca^2+^ conditions, suggesting an influx component in NCa^2+^. On the other hand, UDP elicited a weak response only in NCa^2+^ (Figure 4B). BzATP, a selective agonist of the P2X7 receptor [27], also induced a transient response in both phenotypes in NCa^2+^, but only the response in MFB was abolished in 0 Ca^2+^. This suggests that the transient observed in NCa^2+^ was mediated only by the release from intracellular Ca^2+^ stores (Figure 4C).

Finally, ADP, the specific agonist for the P2Y1 receptor, induced a rapid increment in [Ca^2+^]_i_ that was more robust in qHSC than in MFB. The absence of extracellular Ca^2+^ did not influence the response, suggesting that the response is exclusively regulated by Ca^2+^ release from intracellular stores (Figure 4D). These observations are particularly interesting, given that the P2Y1 receptor has not been identified in qHSC.

### 3.4. ADP-Induced Responses Are Mediated by the P2Y1 Receptor and Depend Exclusively on Release from Intracellular Ca^2+^ Stores

ADP is a potent agonist of the P2Y1 receptor with an EC_50_ of ~250 nM [28]. As shown above (Figure 4D), ADP elicits a response in qHSC and MFB that is strictly dependent on the release from intracellular Ca^2+^ stores. Because the P2Y1 receptor had never been identified in qHSC, we decided to characterize the ADP-dependent responses. First, we constructed concentration–response curves (10 nM to 150 μM) to compare the kinetic characteristics of the responses elicited by ADP in qHSC and MFB. These curves included the magnitude of the response estimated as the area under the curve (AUC) and the number of responding cells (NRC) (Figure 5A,B). The comparison of both parameters indicated a slight rightward shift in the curve and an increment in EC_50_ in MFB (2.9 vs. 5.0 µM for qHSC and MFB, respectively, for AUC and 663.20 nM vs. 1.05 µM for NRCs). This finding suggests that the activation process involves molecular rearrangements of purinergic signaling elements. Particularly, the expression of P2Y1 receptor observed in qHSC is lost in MFB where another pharmacological entity (putatively another receptor) with a different EC_50_ appears.

Subsequently, we confirmed that the response to ADP relies on phospholipase C activity. Cultures of qHSC were incubated for 5 min with 3 µM of U-73122 before stimulation with ADP. The inhibitor completely blocked the effect of 100 nM, 10 µM, and 100 µM of ADP in both qHSC and MFB (Figure 5C,D), further supporting the notion that ADP-dependent responses are exclusively mediated by the release of intracellular Ca^2+^ stores.

Finally, we recorded the ADP-induced responses in the presence of MRS2500, a highly selective antagonist of the P2Y1 receptor [28] (Figure 5E–G). At 2 µM of the antagonist, the effects of ADP at 100 nM, 300 nM, 1µM, and 10 µM were abolished in qHSC. In contrast, the inhibition was only partial at 30, 100 and 150 µM (60.9 ± 5.1, 39.0 ± 5.4 and 29.2 ± 4.9, respectively) (Figure 5E); this suggests the potential existence of a second receptor sensitive to high concentrations of ADP in qHSC. On the other hand, the inhibition exerted by MRS2500 in MFB was marginal (under 10%) for all concentrations (Figure 5F), thereby reinforcing the notion that the ADP-dependent responses in MFB are elicited by a receptor different from the P2Y1 receptor whose function spans the Gq-PLC-IP_3_ pathway.

### 3.5. P2Y1 Receptor Agonism Delays qHSC Activation

HSC activation can be reproduced in vitro by culturing the cells in plastic dishes. Under this condition, the cells remain quiescent for up to 36 h and gradually acquire the MFB phenotype. This process is completed after seven days in culture [14]. To determine whether the P2Y1 receptor plays a role in the activation process, we analyzed the effect of ADP (10 µM), a concentration at which the Ca^2+^ response is abolished by the antagonist MRS2500 (Figure 6), on the phenotype of cultured qHSC after four and seven days, using UV light to quantify fluorescent retinoid vehicles within the cells.

The presence of retinol vesicles serves as a hallmark of the phenotype, and autofluorescence can be observed using phase contrast microscopy (Figure 6A). ADP-treated cells exhibited more retinoid vesicles than control HSC when cultured for four and seven days which indicate a lower degree of transdifferentiation. On day four, the values were 8.1 ± 0.3 vs. 10.5 ± 0.4 vesicles per cell for control and ADP-treated cells, respectively. This difference was more evident after seven days of culture, with 0.3 ± 0.1 vesicles per cell for control cells and 8.9 ± 0.6 vesicles per cell for ADP-treated cells (Figure 6B,C); this depletion of retinol vesicles of non-treated cells on day seven was accompanied by a marked increase in the covered area per cell. To reinforce the notion that P2Y1 receptor is the target of ADP, the described protocol was reproduced preincubating the qHSC with MRS2500 (2 μM) before the stimulus with 10 µM ADP, in these conditions ADP was unable to block the loss of vesicles (Figure 6B). These observations suggest that P2Y1 receptor activity supports the quiescence of HSC and the downregulation of P2Y1 receptor expression facilitates activation.

It is well established that activation of qHSC when cultured on plastic dishes is regulated by mechanotransduction. This mechanism involves the participation of the transcriptional regulators YAP and TAZ [29,30]. To analyze whether ADP delay of qHSC activation is related to regulation of YAP, we evaluated by immunofluorescence the subcellular location of YAP in control or ADP stimulated qHSC at 2 and 7 days of culture, ADP-stimulation induced a reduction in YAP reactivity into the nucleus (Figure 7), suggesting a crosstalk between P2Y1 and YAP.

### 3.6. P2Y1 Receptor Expression Is Downregulated During the Activation of TGF-Stimulated Human qHSC

We analyzed transcriptomic experiments available in the GEO database to assess the expression of the *P2ry1* transcript in human HSC and determine if this expression changes during HSC transdifferentiation. For this analysis, we selected six transcriptomes with data on HSC responses during the activation process. The Methods section details all experimental conditions, accession numbers, and associated publications. The datasets GSE223602, GSE119047, GSE179395, GSE151251, and GSE127964 were obtained from primary cultured HSC derived from healthy donors. GSE232640 was derived from the activated HSC immortalized cell line LX-2. All the studies involved HSC stimulation with TGF-β, a powerful inductor of HSC activation. Data showed that the five transcriptomes from human cells presented negative regulation of *P2ry1* transcript levels upon HSC activation by TGF-β (Figure 8), supporting the notion that the P2Y1 receptor is expressed exclusively by qHSCs and that its function counteracts its transdifferentiation to an MFB phenotype. In contrast, the transcriptome of LX-2 activated HSC exhibited no significant changes in *P2ry1* transcript abundance (Figure 8).

## 4. Discussion

As of October 2025, there have been 11,164 reports published in PubMed regarding HSC. Among these, only 25 reports (i.e., 0.22% of the total), some of which date back to 1994, provide information on purinergic signaling. In addition, several studies used immortalized cell lines derived from HSCs as experimental systems. As a result, publications comparing characteristics and properties of purine receptors in primary cell cultures of qHSC and MFB are extremely scarce.

The in situ emergence of MFB from qHSC is associated with acute or chronic hepatic tissue damage. This complex process is facilitated by diverse cell types and molecular mediators, including interleukins secreted by liver progenitor cells, hepatocytes, Kupffer cells, biliary epithelial cells, and natural killer cells; chemokines produced by pro-fibrotic macrophages, liver sinusoidal endothelial cells, and platelets; and pro-oxidative intermediates formed by hepatocytes, Kupffer cells, and biliary epithelial cells [31]. These cellular signaling events influence the phenotypic transformation of HSCs, but intrinsic factors also promote HSC activation in cultured conditions of purified cells [32].

### 4.1. In Vitro qHSC Transdifferentiation

Elasticity and stiffness in the extracellular milieu, among other factors, are recognized as significant determinants for the adoption of a differentiated phenotype in a given cellular population [33]. The rigidity of the surrounding substrate can modulate cell motility, adhesion, growth, and survival. Therefore, in this study we followed the protocol reported by Dranoff et al. [8] and Bae et al. [34], which involves the transdifferentiation of qHSC to MFB over a seven-day period by cultivating isolated cells in a substrate with high stiffness. Regardless of the use of derivative cell lines, the pro-fibrotic state in isolated qHSC with the concomitant MFB phenotype has also been promoted in vitro by treatment with TGF-β1 [35] and acetaldehyde [36].

After isolation, the HSC phenotype becomes quiescent on day two, and by day seven, it is fully activated. Following transdifferentiation under our experimental conditions, qHSC and MFB presented the typical morphometric and molecular features that characterize each phenotype (Figure 1).

### 4.2. Purinergic Signaling in HSCs

ATP is a well-recognized DAMP [37]. In tissue damage conditions, such as chronic hepatic disease, high concentrations of ATP are released into the extracellular space, mainly from the cytoplasm of dying cells where the concentration of the nucleotide is very high (~10 mM within hepatocytes) [37]. Once in extracellular space, ATP activates the P2X7 receptor, which subsequently triggers the assembly of NLRP3. This inflammasome contributes to the establishment of an inflammatory environment [38]. ATP can also activate other members of the P2X or P2Y purinoceptor family, which exert multiple signaling actions. Thus, in this study, we aimed to demonstrate the expression and functionality of the P2X7 receptor and P2Y receptors coupled with Ca^2+^ mobilization. Recently, it was demonstrated that liver fibrosis was attenuated by inhibiting the release of vesicular ATP in HSC, in both, in vivo and in vitro experiments [39].

Hepatic cell types, similar to those in other organs, express P2Y1, 2, 4, 6 and 11 receptors that act through Gq protein to activate phospholipase C-β, leading to the synthesis of IP_3_ and mobilization of intracellular calcium [40]. Ion channels, such as P2X receptors 1, 2, 3, 4 and 7, can also promote the influx of Ca^2+^; for example, in response to the agonist Bz-ATP [41]. Furthermore, it has been reported that ADO elevates intracellular Ca^2+^ by engaging A2B and A3 receptors through Gq protein stimulation [42].

In the present study, the responses to high concentrations of ATP (100 μM) revealed the functionality of the purinergic system in HSC. This stimulus models the high concentrations of ATP associated with cellular death. The responses exhibited a complex pattern consisting of at least two components: an initial peak dependent on the release from intracellular Ca^2+^ stores and a sustained component mediated by Ca^2+^ influx (Figure 3). The underlying mechanisms and functional implications require further investigation.

RT-PCR amplification and Ca^2+^ recording by functional fluorescence microscopy bring to light the presence and responsiveness of P2Y2, P2Y6, and P2X7 in both qHSC and MFB, corroborating previously published data (Figure 2 and Figure 4) [8]. The pattern of Ca^2+^ mobilization was similar in both cell types in the presence of UDP, UTP, and Bz-ATP, indicating similar activity for P2Y6, P2Y2 and P2X7 receptors. However, the amplitude of the intracellular Ca^2+^ transient was consistently larger in qHSCs across all purinergic responses tested. The *P2ry4* transcript was not amplified, and ADO did not elicit a Ca^2+^ event. Interestingly, the PCR analysis revealed two novel findings: (1) P2Y1 receptor expression in qHSC and (2) the enrichment of P2Y12 receptor in MFB. Therefore, we concentrated on P2Y1 receptor.

The P2Y1 receptor is strongly activated by ADP and is often more responsive to adenine diphosphates than triphosphates; however, ATP has a weak or undetectable effect [43]. The findings of this study support the functional expression of the P2Y1 receptor in qHSC: (1) RT-PCR amplification of a transcript fragment matched with the entry NM_008772.5 in the NCBI database, corresponding to *P2ry1* mRNA; (2) ADP induced potent responses that strictly depended on PLC activity; and (3) the selective antagonist MRS2500 abolished the Ca^2+^ response elicited by 10 µM of ADP in qHSC, a fully activating concentration, and significantly blocked the responses induced by high concentrations (greater than 30 µM). Interestingly, we observed that MRS2500 failed to block the ADP-dependent Ca^2+^ response in MFB (Figure 5B), strongly suggesting that the responsiveness to ADP in this phenotype depends on other receptor(s). This topic requires clarification in new research programs.

To explore the possible role of P2Y1 receptor in qHSC activation, qHSC were cultured on plastic dishes, achieving full activation within seven days. The cells were stimulated with 10 µM of ADP, and activation was analyzed by counting the number of vesicles in comparison to control cells. ADP inhibited vesicle loss and the increment in cellular surface coverage compared with control cells, (Figure 6), which we interpreted as maintenance of quiescence or inhibition of activation. This effect was blocked by MRS2500 indicating the specificity of P2Y1 receptor in the response (Figure 6). In a first approach to describing the mechanisms involved in the P2Y1-dependent actions, the subcellular location of YAP was analyzed in qHSC of MFB stimulated or not with 10 μM of ADP, at this concentration Ca^2+^ mobilization may be completely inhibited by 2 μM of MRS2500. In control cultures an accumulation of YAP in the nucleus was observed at 7 days, ADP induced a decrease in nuclear YAP (Figure 7), suggesting a crosstalk between P2Y1 receptor and YAP protein and reveals a mechanism displayed by P2Y1 on mechanotransduction pathways.

These results strongly suggest that ADP accumulation represents an autocrine-paracrine signal that sustains quiescence. It is clear that the enrichment of ADP in the extracellular space is a result of enzyme-assisted dephosphorylation of ATP. In this context, investigating the functional expression of ectonucleotidases by HSCs could be valuable. Despite reports on the expression of NTPDase1 (CD39), NTPDase 2 [8], CD73 [44], and NPP2 and 3 [45], a detailed study of nucleotide dynamics in HSCs and HSC activation is still necessary.

The actions described of the P2Y1 receptor may have translational implications. To explore whether parallel mechanisms operate in human HSC, we analyzed transcriptomes from the GEO database. Five of the six transcriptomes were constructed from primary human HSC and one was obtained from the activated HSC immortalized cell line LX-2; HSC activation was induced by the administration of TGF-β. In agreement with our observations, *P2RY1* transcript expression was consistently downregulated across all datasets from primary-cultured HSC, indicating that P2Y1 receptor expression is a marker for the quiescent phenotype (Figure 8). This finding opens the door to developing translational research programs to evaluate the feasibility of P2Y1 receptor as a phenotypic marker and molecular target in the mechanism underlying HSC activation and the associated fibrotic response.

## 5. Conclusions

The experimental evidence in the present study demonstrates the functional expression of P2Y receptors coupled to Ca^2+^ mobilization in mouse qHSC and MFB. The functional expression of P2Y1, P2Y2 and P2Y6, and additionally P2X7 receptor was confirmed. Of special interest was the unpublished role played by the P2Y1 receptor in the activation process of qHSC. The activation of this receptor by ADP (10 μM) ameliorated the formation of MFB. At this concentration of ADP, the P2Y1 receptor antagonist MRS2500, abolished the ADP-dependent Ca^2+^ mobilization, suggesting that the predominant role played by the P2Y1 receptor. Finally, experimental data suggested that ADP-dependent delay of qHSC activation is mediated by the inhibition of the nuclear translocation of YAP, a master transcriptional regulator of mechanotransduction. Hence, we are contributing to the little explored topic of purinergic regulation of HSC, functional differences between quiescent and activated cells that could be outstanding during the transdifferentiation process with potential consequences in the installation of the liver fibrotic condition.

## Figures and Tables

**Figure 1 cells-14-01845-f001:**
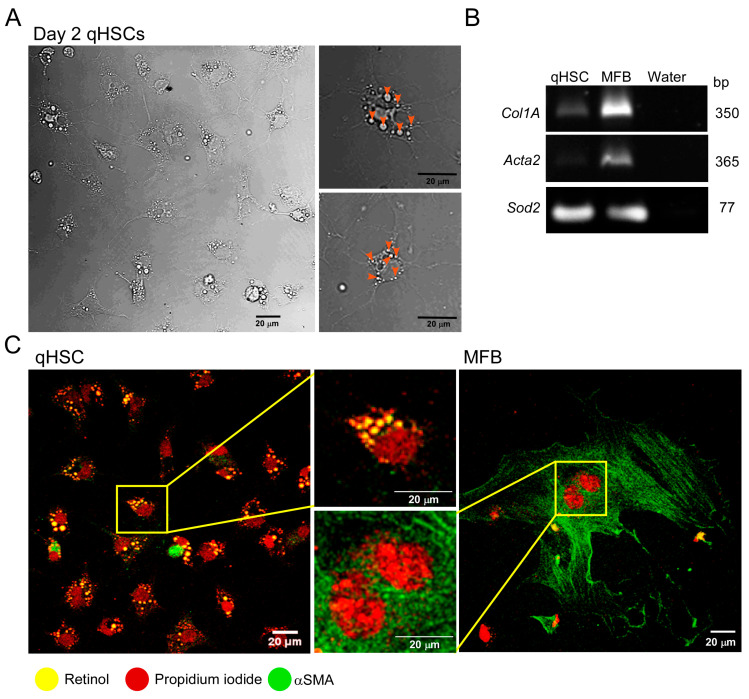
Primary cultured qHSC transdifferentiate to MFB. (**A**) Phase contrast image of a typical culture of qHSC on day two which shows the high purity of the preparation. Amplified single cells are depicted on the right, with retinoid vesicles clearly distinguishable (orange arrowheads). (**B**) End-point PCR showing the amplicons for *Col1A1* and *Acta2* as activation markers and *Sod2* serving as a housekeeping transcript. (**C**) Immunolabeling of αSMA in qHSC and MFB (green signal). Nuclei were stained with propidium iodide (red signal). Retinol vesicles exhibit a yellow fluorescence when excited with blue light. The RNAs used for RT-PCR were from a pooled sample of three cultures obtained from different mice.

**Figure 2 cells-14-01845-f002:**
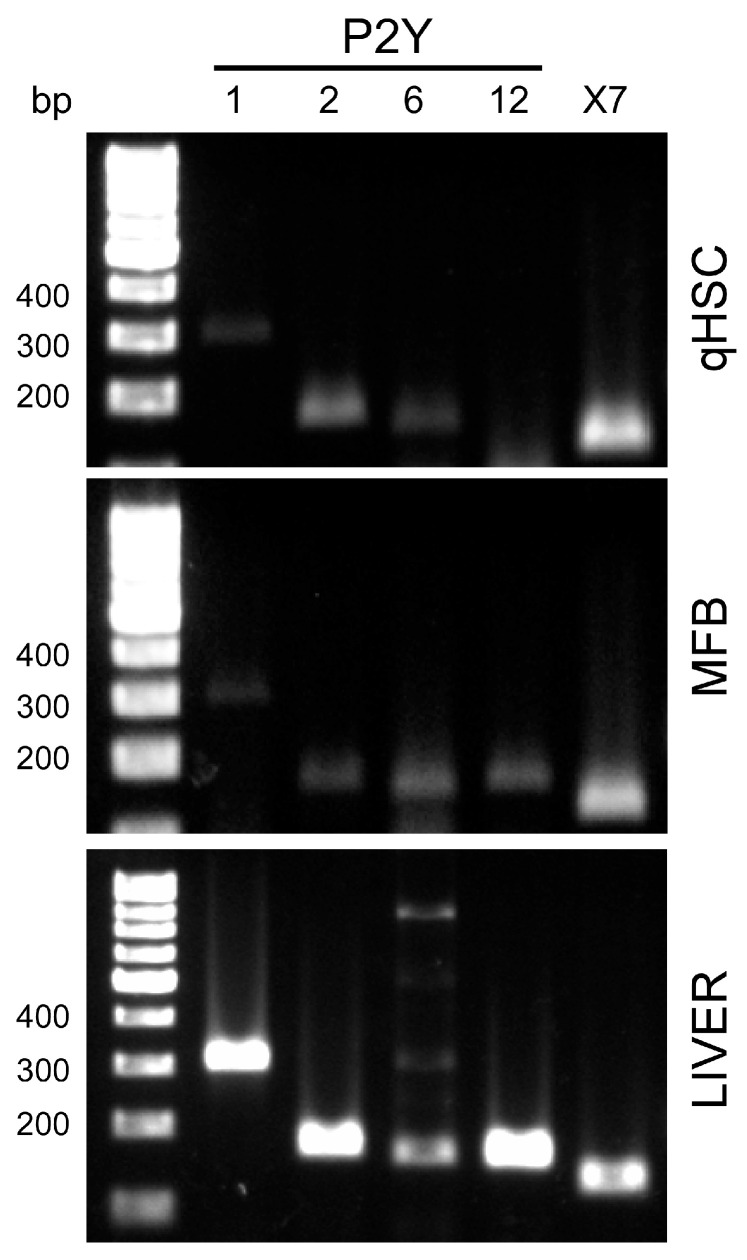
Detection of purinergic receptor transcripts in qHSC and MFB. cDNA was synthesized from total RNA from qHSC and MFB and used for PCR amplification of *P2ry1* (315 bp), *P2ry2* (174 bp), *P2ry6* (152 bp), *P2ry12* (162 bp), and *P2rx7* (129 bp). Representative images of amplicons analyzed in 1% agarose gels are shown. Liver homogenate cDNA was used as a positive control. All amplicons were purified and sequenced, and their identity was confirmed using the BLAST platform.

**Figure 3 cells-14-01845-f003:**
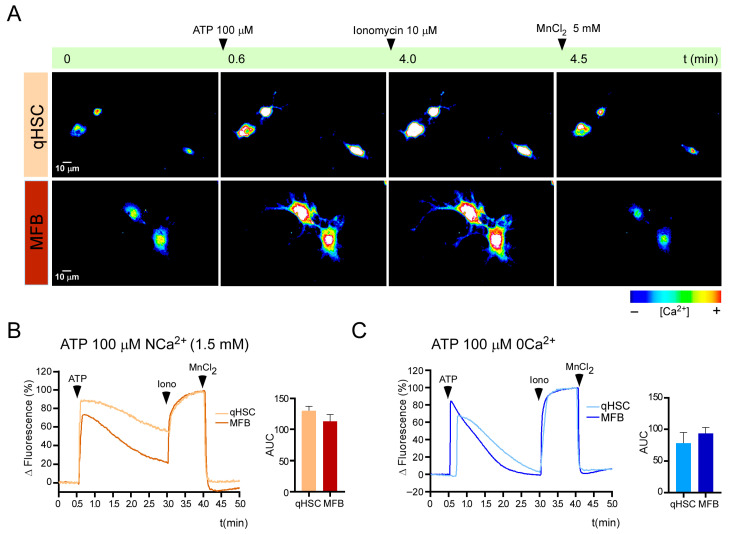
ATP elicited Ca^2+^ responses in qHSC and MFB. (**A**) Sequence of fluorescent images from qHSC or MFB loaded with Fluo-4 AM (2 µM) showing changes in [Ca^2+^]_i_ before and after the addition of 100 µM ATP (arrowhead) in NCa^2+^ extracellular solution. Pseudocolor from black to red represents low to high [Ca^2+^]_i_, respectively; timeframes are indicated in minutes. (**B**,**C**) show representative traces (left panel) and AUC quantification (right panel; mean ± s.e.m. of the AUC of the Ca^2+^ transient) of cells stimulated with 100 μM ATP in NCa^2+^ (**B**) or 0 Ca^2+^ (**C**) extracellular solutions. At the end of the protocol, ionomycin (10 µM) and MnCl_2_ (5 mM) were sequentially applied to determine the maximum and minimum levels of intracellular Ca^2+^, respectively. When ionomycin was added under 0 Ca^2+^ conditions, extracellular Ca^2+^ was restored to levels of NCa^2+^. At least 50 cells were analyzed per experiment, n = 3.

**Figure 4 cells-14-01845-f004:**
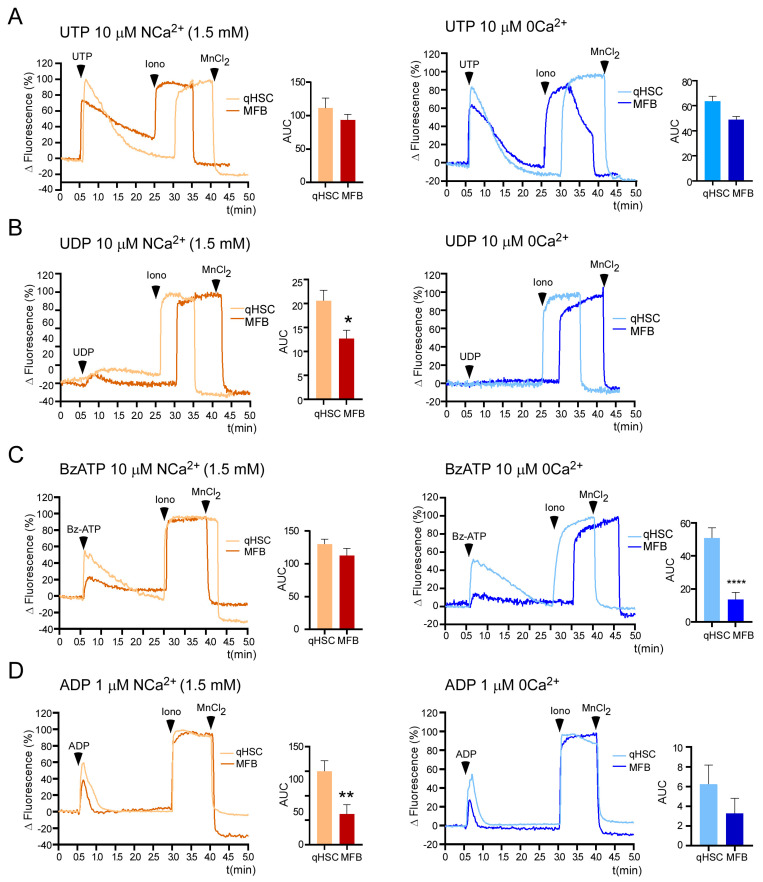
Purinergic agonists elicited Ca^2+^ responses in qHSC and MFB. qHSC and MFB were stimulated with purinergic agonists in normal (NCa^2+^, 1.5 mM Ca^2+^) or calcium-free (0 Ca^2+^) Krebs solution. Representative traces (left panels) and AUC quantification (right panels; mean ± s.e.m. of the AUC of the Ca^2+^ transient) of cells stimulated in NCa^2+^ or 0 Ca^2+^ with 10 µM UTP (**A**), 10 µM UDP (**B**), 50 µM BzATP (**C**) and 1 µM of ADP (**D**). At the end of the protocol, ionomycin (10 µM) and MnCl_2_ (5 mM) were sequentially applied to determine the maximum and minimum levels of intracellular Ca^2+^, respectively. When ionomycin was added under 0 Ca^2+^ conditions, extracellular Ca^2+^ was restored to levels observed in NCa^2+^. At least 50 cells were analyzed per experiment, n = 3. * *p* < 0.05, ** *p* < 0.01 and **** *p* < 0.0001, Mann–Whitney test.

**Figure 5 cells-14-01845-f005:**
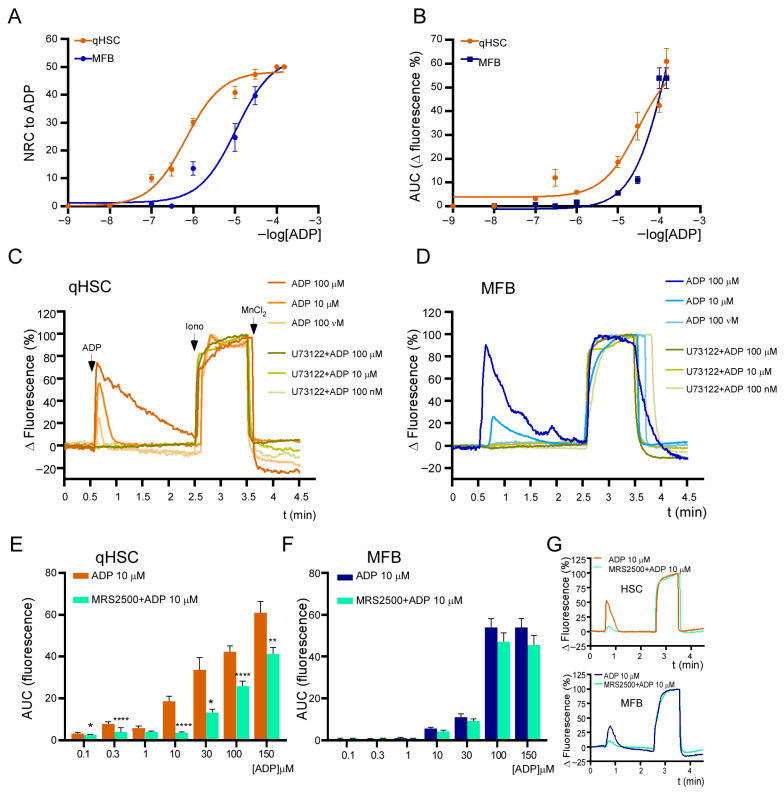
Calcium responses elicited by ADP in qHSC and MFB. Concentration–response curves in cultures of qHSC and MFB with concentrations of ADP ranging from 10 nM to 150 µM of ADP. The curves depict the number of responding cells (NRCs) (**A**) and area under the curve (AUC) (**B**). Representative traces showing the effect of 3 µM of the PLC inhibitor U73122 on the Ca^2+^ responses elicited by ADP (100 nM, 10 µM and 100 µM) in qHSC (**C**) and MFB (**D**). Effect of MRS2500, a selective antagonist of the P2Y1 receptor (2 µM), on the Ca^2+^ responses elicited by ADP (100 nM to 150 µM) in qHSC (**E**) and MFB (**F**). Representative traces of ADP at 10 µM in HSCs (upper panel) and MFBs (lower panel) (**G**). At least 50 cells were analyzed per experiment, n = 3. * *p* < 0.05, ** *p* < 0.01, **** *p* < 0.0001, Mann–Whitney test.

**Figure 6 cells-14-01845-f006:**
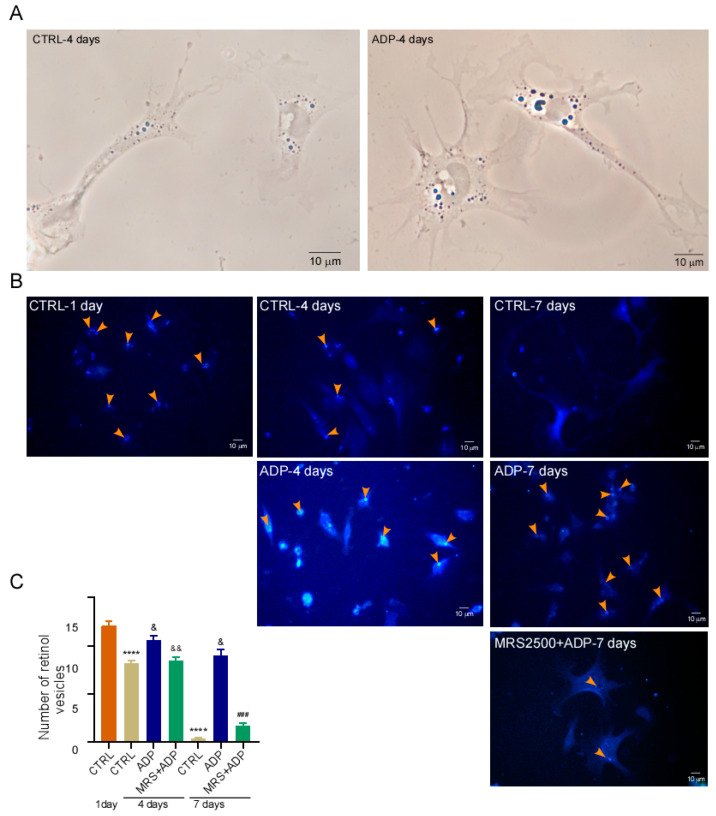
P2Y1 receptor activation delays the transdifferentiation of qHSC into MFB. (**A**) Representative images of HSC cultured for 4 days without (**left**) and with ADP 10 µM (**right**) visualized using phase-contrast microscopy. Retinoid vesicles are clearly distinguishable (blue droplets). (**B**) Representative retinoid fluorescence images (autofluorescence produced by UV light excitation) showing lipid droplets (orange arrows) in HSC cultured with and without ADP 10 µM in presence or absence of MRS2500 (2 µM) by the indicated time, the retinol vesicle quantification (mean ± s.e.m.) is showed in (**C**). At least 50 cells were analyzed per experiment, n = 3. **** *p* < 0.0001 vs. CTRL 1 day; ^&^ *p* < 0. 05 vs. CTRL 1 day; ^&&^
*p* < 0.01 vs. CTRL 4 days; ^###^ *p* < 0.0001 vs. CTRL 7 days. Kruskal–Walli’s test.

**Figure 7 cells-14-01845-f007:**
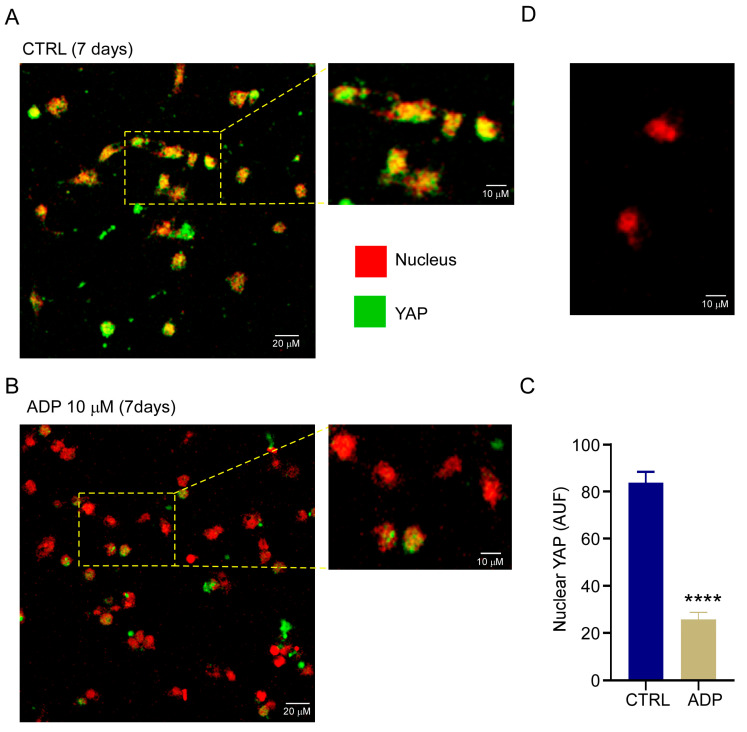
ADP counteracts the nuclear translocation of YAP in the activation of HSC. Isolated mouse HSC were cultured on plastic dishes for 7 days in control conditions (**A**) or in presence of 10 µM ADP (**B**). Then, YAP was detected by immunofluorescence and intensity of fluorescent signal into the nucleus quantified (**C**). Samples where the primary antibody was omitted did not present any detectable green signal (**D**). **** *p* < 0. 0001. Kruskal–Walli’s test.

**Figure 8 cells-14-01845-f008:**
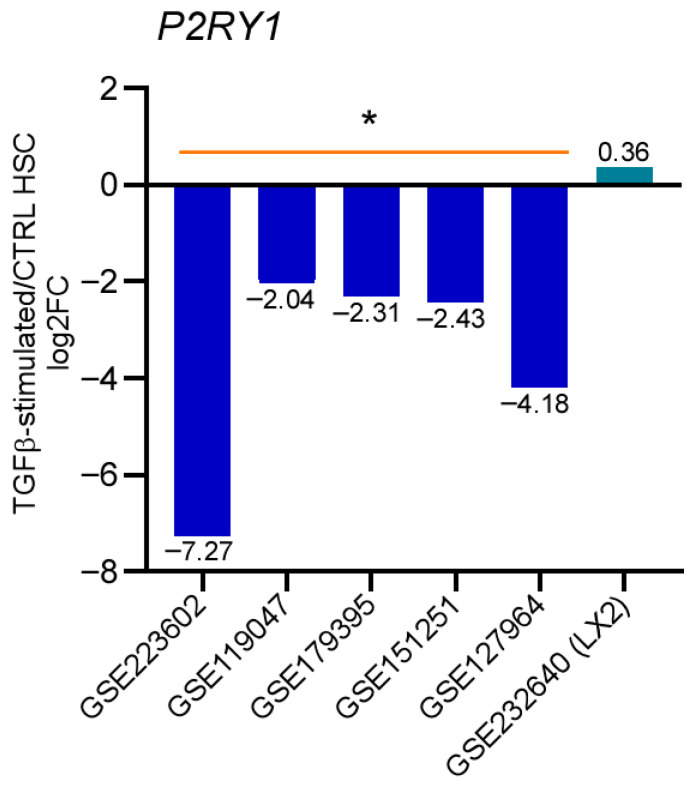
Regulation of *P2ry1* expression in healthy qHSC and after activation by TGF-β. Transcriptomic profiles from independent studies were obtained from Gene Expression Omnibus (GEO) [16]. In all the datasets, MFB were compared to qHSC using the GEO2R tool, [17] as previously described. Table 2 outlines all datasets containing human HSCs activated by TGF-β and the specific conditions of each experiment. Data are presented as means of log2 Fold Change (log2FC), * *p*-value < 0.05.

**Table 1 cells-14-01845-t001:** Oligonucleotides used in the study.

Target Transcript	Forward	Reverse
*Acta2*	CTGAGCGTGGCTATTCCTTC	CTTCTGCATCCTGTCAGCAA
*Col1A1*	GAGCGGAGAGTACTGGATCG	CCTTCTTGAGGTTGCCAGTC
*P2ry1*	TACCAGCCCTCATCTTCTAC	CATTGGACGTGGTGTCATAG
*P2ry2*	ACCTGGAACCCTGGAATAG	AGGCGGCATAGGAAGATATAG
*P2ry4*	CCTGGACTGGACTAAGGAA	TCAGAGGCAACAGGATGA
*P2ry6*	TCTGGCACTTCCTCCTAAA	CTTGAAATCCTCACGGTAGAC
*P2ry12*	CAGTCTGCAAGTTCCACTAAC	TGGGTGATCTTGTAGTCTCTG
*P2rx7*	TGACGAAGTTAGGACACAGC	GGATACTCAGGACACAGCG
*Sod2*	TGGACAAACCTGAGCCCTAA	GACCCAAAGTCACGCTTGATA

**Table 2 cells-14-01845-t002:** Overview of transcriptomes analyzed from the GEO database.

Accession Number	Experimental Conditions	Control	Reference
GSE223602	3× healthy derived donor TGFB (5 ng/mL) for 24 h	3× healthy derived donor ctrl	[19]
GSE119047	3× healthy derived donor TGFB (2.5 ng/mL) for 24 h	3× healthy derived donor ctrl	[20]
GSE179395	8× healthy derived donor TGFB (2 ng/mL) for 96 h	6× healthy derived donor ctrl	[21]
GSE151251	3× healthy derived donor TGFB	3× healthy derived donor ctrl	
GSE127964	3× healthy derived donor TGFB (5 ng/mL) for 24 h	3× healthy derived donor ctrl	[22]
GSE119606	3× healthy derived donor TGFB (10 ng/mL) for 48 h	3× healthy derived donor ctrl	[23]
GSE232640	4× LX2 cell line TGFB (5 ng/mL) for 16 h	4× LX2 cell line ctrl	[24]

## Data Availability

Data supporting this study are available from the corresponding author on reasonable request.

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
