# Peer review of "Purinergic-Mediated Calcium Signaling in Quiescent and Activated Hepatic Stellate Cells: Evidence That P2Y1 Receptor Delays Activation"

_cells, 2025, doi:10.3390/cells14231845_

Round 1
Reviewer 1 Report
Comments and Suggestions for Authors
In the present study, Esperanza Mata-Martínez et. al. focused on purinergic receptor-mediated calcium signaling in hepatic stellate cells (HSCs). They identified functional P2Y1 receptors in quiescent hepatic stellate cells and confirmed that ADP, the agonist of P2Y1 receptors, can delay the transformation of qHSCs into myofibroblasts. This finding provides a new target for the study of hepatic fibrosis mechanisms and holds certain scientific value. Despite the novelty of P2Y1 receptors, there are some concerns that should be considered.
Major Concerns:
- The paper lacks further mechanism research. It only focuses on the "existence-function" correlation of P2Y1 receptors, but lacks the molecular mechanism by which P2Y1 receptors regulate HSC transformation. For example, it does not clarify the interaction between P2Y1 receptors and known fibrosis signaling pathways (such as the TGF-β/Smad pathway), nor does it analyze the specific downstream effector molecules of calcium signaling.
- The paper confirms that the calcium signaling mediated by P2Y1 receptors depends on PLC activity, but does not analyze the impact of this calcium signaling on other activation-related biological processes of HSCs, such as proliferation and apoptosis.
Minor Concerns:
- Fig 1 Some images lack a scale bar.
2.English expression needs improvement.
Comments on the Quality of English Language1) "MFB" (myofibroblast) and "MBF" are used interchangeably throughout the paper. For example, in Section 3.3, it is stated: "Figure 3A presents a representative series of images illustrating a response elicited by 100 μM of ATP in qHSC and MFB in NCa²⁺ conditions"; however, "MBF" is incorrectly used in some subsequent paragraphs.
2) "P2ry" and "P2Y" exhibit inconsistent capitalization (e.g., "p2ry1" and "P2Y1" alternate within the same paragraph), which is likely to cause confusion.
3) In Section 3.4, the sentence "the expression of P2Y1 receptor observed in qHSC is lost in MFB where another pharmacological entity (putatively another receptor) with a different EC50 appears" contains an incomplete structure. The attributive clause introduced by "where" lacks a predicate verb.
Author Response
Reviewer 1
In the present study, Esperanza Mata-Martínez et. al. focused on purinergic receptor-mediated calcium signaling in hepatic stellate cells (HSCs). They identified functional P2Y1 receptors in quiescent hepatic stellate cells and confirmed that ADP, the agonist of P2Y1 receptors, can delay the transformation of qHSCs into myofibroblasts. This finding provides a new target for the study of hepatic fibrosis mechanisms and holds certain scientific value. Despite the novelty of P2Y1 receptors, there are some concerns that should be considered.
Major Concerns:
Comment 1. The paper lacks further mechanism research. It only focuses on the "existence-function" correlation of P2Y1 receptors, but lacks the molecular mechanism by which P2Y1 receptors regulate HSC transformation. For example, it does not clarify the interaction between P2Y1 receptors and known fibrosis signaling pathways (such as the TGF-β/Smad pathway), nor does it analyze the specific downstream effector molecules of calcium signaling.
Response 1. The reviewer´s point of view is pertinent, but this study was originally designed to characterize the expression and the functional calcium responses of all the Gq-coupled P2Y receptors in HSC. As the reviewer notes, one of the main findings is the expression of P2Y1 receptor in quiescent phenotype and that its stimulation with ADP delays the activation process. However, in the present manuscript we are just describing the findings of the phenomenon.
However, in an effort to initiate the initial description of the mechanisms involved in the ADP-dependent delay of MFB activation, we evaluated the effect of ADP on the nuclear translocation of the YAP, a transcriptional regulator related with mechanotransduction, because, in the present study HSC activation is mainly promoted by substrate stiffness (Mennaerts et al., doi:10.1016/j.jhep.2015.04.011).
The results showed that in control HSC cultured by 7 days in plastic dishes, YAP is located into the nucleus. The incubation with 10 mM of ADP decreased the levels of nuclear YAP. These results are included in the new Figure 7, lines 378-384.
Comment 2. The paper confirms that the calcium signaling mediated by P2Y1 receptors depends on PLC activity but does not analyze the impact of this calcium signaling on other activation-related biological processes of HSCs, such as proliferation and apoptosis.
Response 2. These aspects are of broad interest but are part of a subsequent project.
Minor Concerns:
1.-Fig 1 Some images lack a scale bar.
The mistake was fixed
2.-English expression needs improvement.
Comments on the Quality of English Language:
1)"MFB" (myofibroblast) and "MBF" are used interchangeably throughout the paper. For example, in Section 3.3, it is stated: "Figure 3A presents a representative series of images illustrating a response elicited by 100 μM of ATP in qHSC and MFB in NCa²⁺ conditions"; however, "MBF" is incorrectly used in some subsequent paragraphs.
The mistakes were fixed, and the manuscript was corrected by a native English speaker professional in proofreading.
2) "P2ry" and "P2Y" exhibit inconsistent capitalization (e.g., "p2ry1" and "P2Y1" alternate within the same paragraph), which is likely to cause confusion.
All the P2Y1 receptor nomenclature, specifically for gene and for protein designation, was corrected.
3) In Section 3.4, the sentence "the expression of P2Y1 receptor observed in qHSC is lost in MFB where another pharmacological entity (putatively another receptor) with a different EC50 appears" contains an incomplete structure. The attributive clause introduced by "where" lacks a predicate verb.
The mistake was fixed, and the manuscript was corrected by a native English speaker professional in proofreading.
Reviewer 2 Report
Comments and Suggestions for Authors
In the present work, Mata-Martínez et al. try to compare the expression of purinergic receptors with the capacity to mobilize intracellular Ca2+ in both phenotypes, as well as to explore the potential role of these signals in hepatic stellate cell activation. However, there are some questions that should be answered.
Major concerns
- The manuscript mentions analyzing public GEO datasets but does not include a formal data availability statement. In addition, public GEO datasets used in this manuscript are involved in humans, were which approved by the Institute of Neurobiology’s Bioethics Committee at the National Autonomous University of Mexico (UNAM) (protocol 85.A) and comply with Official Mexican Standard SAGARPA NOM-062-ZOO-1999?
- Hepatic stellate cells should be verified, but there is no related information.
- Lack of direct functional link to P2Y1 in activation assay. While the study convincingly shows that ADP elicits a Ca2+ response via P2Y1 in qHSCs, the experiment in Section 3.5 (Figure 6) only tests the effect of ADP on activation, not the specific role of the P2Y1 receptor.
- Figure 2 is inconsistent with the original image using black and white reversal. This is a misconception that it is a WB.
- English grammar and writing style should be checked and revised throughout the manuscript.
Minor concerns
- Revise the Highlights based on the Cells style, and there should be no serial number.
- Keywords, revise ‘Ito cells’.
- Line 63, delete ‘(PPARγ)’ and ‘(GFAP)’. If the abbreviation only appears once, please delete it, and check and revise it throughout the manuscript.
- Line 80, the sentence "In 2004, Rebeca Wells and colleagues published the first report..." should be cited properly. The current citation [7] is for a 2004 paper by Dranoff et al., not Rebeca Wells. Please verify and correct the reference.
- Lines 114-117 and Lines 523-525, ‘All experimental procedures were….’, there are repeats.
- Lines 141-152, a table may be used for these oligonucleotides sequences.
- Line 167, revise format ‘= 2.4 ? 10−11 M;’.
- Table 1, correct ‘5ng/mL/24h’.
- Line 238, ‘P2ry1, P2ry2, P2ry4, and P2ry6’, all gene abbreviation should be in italic. Please check and revise it throughout the manuscript, including the figure legends.
- Line 345, correct "MBF".
- Line 365, revise "ADP treatment prevented the expression of αSMA and Col1a1 in MFB" to "ADP treatment prevented the upregulation of αSMA and Col1a1 during HSC activation".
- Line 412, ‘p-value’, ‘p’ should be in italic. Please check and revise it throughout the manuscript,
- References section. Ref.13, delete ‘(Basel)’.
The English could be improved to more clearly express the research.
Author Response
Reviewer 2
In the present work, Mata-Martínez et al. try to compare the expression of purinergic receptors with the capacity to mobilize intracellular Ca2+ in both phenotypes, as well as to explore the potential role of these signals in hepatic stellate cell activation. However, there are some questions that should be answered.
Major concerns
Comment 1. The manuscript mentions analyzing public GEO datasets but does not include a formal data availability statement. In addition, public GEO datasets used in this manuscript are involved in humans, were which approved by the Institute of Neurobiology’s Bioethics Committee at the National Autonomous University of Mexico (UNAM) (protocol 85.A) and comply with Official Mexican Standard SAGARPA NOM-062-ZOO-1999?
Response 1. Gene Expression Omnibus is an international public repository that archives and freely distributes array, next-generation sequencing and other functional genomics data. The transcriptomes analyzed are public and available for everybody, have a unique ID and are linked to an article. The ID and the article associated with each transcriptome used in the present study may be consulted in Table 1. Copyright status can be consulted in : https://www.ncbi.nlm.nih.gov/geo/info/disclaimer.html
The performance of a transcriptomic study must comply with international bioethical parameters, which are applied by each research institution and corroborated through the publication process by each publisher. The use of GEO-Omnibus data does not require local bioethical approval.
Comment 2. Hepatic stellate cells should be verified, but there is no related information.
Response 2. The most accepted characteristic to verify the identity of HSC is the presence of retinol vesicles; and, for the activated phenotype the increment on Col1a1 and a-SMA, Figure 1 shows the most accepted markers for qHSC and MFB.
Comment 3. Lack of direct functional link to P2Y1 in activation assay. While the study convincingly shows that ADP elicits a Ca2+ response via P2Y1 in qHSCs, the experiment in Section 3.5 (Figure 6) only tests the effect of ADP on activation, not the specific role of the P2Y1 receptor.
Response 3. To highlight the specific contribution of P2Y1 receptor in the ADP-dependent delay of qHSC activation, it was assayed the effect of MRS2500, a potent antagonist of P2Y1 receptor, before the stimulus with ADP. MRS2500 significantly blocked the effect of ADP, supporting the importance of P2Y1 receptor in the MFB activation, these data were included in the new Figure 6, lines 363-366.
Comment 4. Figure 2 is inconsistent with the original image using black and white reversal. This is a misconception that it is a WB.
Response 4. In the corrected version, images in Figure 2 are present in their original version.
Comment 5. English grammar and writing style should be checked and revised throughout the manuscript.
Response 5. The manuscript was corrected by a native English speaker professional in proofreading.
Minor concerns
- Revise the Highlights based on the Cells style, and there should be no serial number.
Done
- Keywords, revise ‘Ito cells’.
It is a synonym for HSC
- Line 63, delete ‘(PPARγ)’ and ‘(GFAP)’. If the abbreviation only appears once, please delete it, and check and revise it throughout the manuscript.
Abbreviations were deleted
- Line 80, the sentence "In 2004, Rebeca Wells and colleagues published the first report..." should be cited properly. The current citation [7] is for a 2004 paper by Dranoff et al., not Rebeca Wells. Please verify and correct the reference.
Rebeca Wells was the PI of this group; however, the text was modified to avoid confusion.
- Lines 114-117 and Lines 523-525, ‘All experimental procedures were….’, there are repeats.
The mistake was fixed
- Lines 141-152, a table may be used for these oligonucleotides sequences.
A Table containing oligonucleotide sequences was added
- Line 167, revise format ‘= 2.4 ? 10−11 M;’.
Done
- Table 1, correct ‘5ng/mL/24h’.
Done
- Line 238, ‘P2ry1, P2ry2, P2ry4, and P2ry6’, all gene abbreviation should be in italic. Please check and revise it throughout the manuscript, including the figure legends.
Done
- Line 345, correct "MBF".
Done
- Line 365, revise "ADP treatment prevented the expression of αSMA and Col1a1 in MFB" to "ADP treatment prevented the upregulation of αSMA and Col1a1 during HSC activation".
Done
- Line 412, ‘p-value’, ‘p’ should be in italic. Please check and revise it throughout the manuscript,
Done
- References section. Ref.13, delete ‘(Basel)’.
Done
Reviewer 3 Report
Comments and Suggestions for Authors
Purinergic-mediated calcium signaling in quiescent and activated hepatic stellate cells: Evidence that P2Y1 receptor delays activation
The title refers to a study showing that P2Y1 purinergic receptors delay the activation of hepatic stellate cells (HSCs) by modulating calcium signaling. In both quiescent and activated states, these receptors are involved in regulating intracellular calcium levels, which play a crucial role in HSC function. Standard protocol for experimental studies was approved by the Institute of Neurobiology’s Bioethics Committee at the National Autonomous University of México(UNAM) (protocol 85.A) and comply with Official Mexican Standard SAGARPA NOM 062-ZOO-1999. The animal management protocol was designed to inflict as little pain as possible. Animals were kept under control conditions (23°C, a 12/12-h light-dark cycle, and food and water ad libitum). Study design is well done. Methods used are well chosen. Authors used Reverse transcription and PCR, for intracellular calcium measurement fluorescent microcopy and for P2RY1 transcript expression public databases. Results were presented in 3 figures which depict fluorescent images and figure 4 and 5 presenting purinergic agonits and calcium responses elicited by ADP. References and introduction must be up-dated
Purinergic-mediated calcium signaling in quiescent and activated hepatic stellate cells: Evidence that P2Y1 receptor delays activation
The title refers to a study showing that P2Y1 purinergic receptors delay the activation of hepatic stellate cells (HSCs) by modulating calcium signaling. In both quiescent and activated states, these receptors are involved in regulating intracellular calcium levels, which play a crucial role in HSC function. Standard protocol for experimental studies was approved by the Institute of Neurobiology’s Bioethics Committee at the National Autonomous University of México(UNAM) (protocol 85.A) and comply with Official Mexican Standard SAGARPA NOM 062-ZOO-1999. The animal management protocol was designed to inflict as little pain as possible. Animals were kept under control conditions (23°C, a 12/12-h light-dark cycle, and food and water ad libitum). Study design is well done. Methods used are well chosen. Authors used Reverse transcription and PCR, for intracellular calcium measurement fluorescent microcopy and for P2RY1 transcript expression public databases. Results were presented in 3 figures which depict fluorescent images and figure 4 and 5 presenting purinergic agonits and calcium responses elicited by ADP. References and introduction must be up-dated
Author Response
Purinergic-mediated calcium signaling in quiescent and activated hepatic stellate cells: Evidence that P2Y1 receptor delays activation
Comment 1. The title refers to a study showing that P2Y1 purinergic receptors delay the activation of hepatic stellate cells (HSCs) by modulating calcium signaling. In both quiescent and activated states, these receptors are involved in regulating intracellular calcium levels, which play a crucial role in HSC function. Standard protocol for experimental studies was approved by the Institute of Neurobiology’s Bioethics Committee at the National Autonomous University of México(UNAM) (protocol 85.A) and comply with Official Mexican Standard SAGARPA NOM 062-ZOO-1999. The animal management protocol was designed to inflict as little pain as possible. Animals were kept under control conditions (23°C, a 12/12-h light-dark cycle, and food and water ad libitum). Study design is well done. Methods used are well chosen. Authors used Reverse transcription and PCR, for intracellular calcium measurement fluorescent microcopy and for P2RY1 transcript expression public databases. Results were presented in 3 figures which depict fluorescent images and figure 4 and 5 presenting purinergic agonits and calcium responses elicited by ADP. References and introduction must be up-dated.
Response 1. Two new references were included in the Introduction.
Round 2
Reviewer 1 Report
Comments and Suggestions for Authors
It is acceptable now.
Author Response
Thanks for your help
Reviewer 2 Report
Comments and Suggestions for Authors
Thanks for author’s responses. However, there are some minor concerns.
- The subsection titles of Materials and Methods and Discussion sections should be numbered.
- Check the format of ‘3.6. P2Y1 receptor expression….’ (Lines 392-393).
- Check the format of Ref. 38 and 39 (Lines 631-635).
Author Response
Reviewer 2
- The subsection titles of Materials and Methods and Discussion sections should be numbered.
Done
- Check the format of ‘3.6. P2Y1 receptor expression….’ (Lines 392-393).
Done
- Check the format of Ref. 38 and 39 (Lines 631-635).
Done